# Gender biases and hate speech: Promoters and targets in the Argentinean political context

Laia Domenech Burin[1,2]*, Juan Manuel Pérez[3], Germán Rosati[1,4], Magalí Rodrigues Pires[5], María Nanton[5], Diego Kozlowski[6]

1 Escuela Interdisciplinaria de Altos Estudios Sociales, Universidad de San Martín, Buenos Aires, Argentina, 2 Department of Statistics, Social Data Science and AI Lab (SODA), LMU Munich, Munich, Germany, 3 Facultad de Ciencias Exactas y Naturales, Instituto de Ciencias de la Computación, Universidad de Buenos Aires, Buenos Aires, Argentina, 4 Consejo Nacional de Investigaciones Científicas y Técnicas, Buenos Aires, Argentina, 5 Facultad de Ciencias Exactas y Naturales, Maestría en Data Mining & Knowledge Discovery, Universidad de Buenos Aires, Buenos Aires, Argentina, 6 École de Bibliothéconomie et des Sciences de l'information, Université de Montréal, 3150 rue Jean-Brillant, Montréal, QC, Canada

* ldomenechburin@unsam.edu.ar

**Data Availability Statement:** All complete dataset files are available from the OSF repository https://osf.io/xqd3g/?view_only=9c6abb016f01449899ff4435616802e3.

## Abstract

Hate speech found in social media a place to flourish. In the Argentinean context, new right-wing parties have disrupted the political arena, winning the elections of 2023. Many of these new right-wing figures grew in popularity due to their use of social media, on a background of increasing political violence. In this article, we use quantitative and qualitative tools to investigate the prevalence of hate speech targeting women politicians and analyze the role of different political affiliations in promoting such discourse. Furthermore, we propose a model that predicts users' political alignments based on their profile descriptions, allowing us to explore the distribution of hate speech among different political orientations. Our results provide a descriptive account of the relationship between hate speech by politicians and other users and shows that right-wing political figures and supporters are strong emissors of hate speech, while women, especially those from the left-wing are more prone to receive violent content.

## 1. Introduction

In recent times, hate speech has emerged as a relevant issue due to its negative effects on a societal level. It directly impacts the recipients of such rhetoric [1] and contributes to the radicalization of political stances, leading to physical violence against marginalized groups [2–4]. Consequently, international organizations like the United Nations have devised strategies and protocols to combat and eradicate this form of violence [5].

Hate speech can be described as any speech in the public sphere that aims to promote, incite, or legitimize discrimination, dehumanization, and violence against individuals or groups based on their religious, ethnic, national, political, racial, gender, or other social identity [6]. While hate speech has a long history, modern communication technologies have

**Funding:** The author(s) received no specific funding for this work.

**Competing interests:** The authors have declared that no competing interests exist.

expanded its impact. Social media has become the prime platform for the spread and amplification of such discourse, leading to various studies focused on detecting hate speech on platforms like Twitter [5,7,8]. These speeches may also serve political purposes. Díez Gutierrez et al. [9] have studied how far-right groups employ hate speech as a tactic to advance campaigns against women's and marginalized groups' rights while discrediting public figures. Women, and especially feminist activists, are a frequent target of this type of online discourse. This kind of violence is termed digital violence, referring to gender-based violence enabled or exacerbated by Information and Communication Technologies (ICT) [10].

Far-right groups and users engage in delegitimization and violence campaigns targeting women activists. Chaher [11], reveals how these groups present themselves in public debates through disqualifying, aggressive, or violent means. Feminist activists are frequently singled out by such groups when they express their views and take positions in public discussions on social media.

Conversely, there is a strong correlation between the dissemination of hate speech and the rise of certain right-wing political figures on social media platforms. The establishment of political factions that discredit and target various social movements fosters the amplification of hate speech, granting it credibility and social acceptance [12]. Even though hate speech is a global issue, it has a strong contextual element. At each region, the promoters, the targets, and the content of the hate speech messages may vary. Within the Latin American region, the recent rise in power of a figure like Javier Milei, who became first known as a twitter and media figure, poses the question on the specific forms that hate speech takes in Argentina's social media. New developments in hate speech detection for rioplatense Spanish [13] present an opportunity for new quantitative approaches.

Currently, there is a lack of research on the circulation of hate speech targeted towards women in the Argentine political scenario, particularly using computational tools. Although there are studies that either study the biased, misogynistic nature of hate speech, its role as a tool for attacking women in politics, or its connection to the rise of right-wing parties, there is no research that integrates these three topics using computational methods. This work aims to address this gap through three main research questions. First, is there a gender bias in the circulation of hate speech related with Argentinean political figures? Secondly, which political figures are responsible for promoting this form of digital violence? And lastly, which groups replicate this violence?

The following section provides the theoretical framework to guide the analysis. Section 3 describes related works on this topic and how this work differs from the existing ones. Section 4 presents the data and methods used. Section 5 shows the results. We start by measuring the proportion of emitted and received hate speech for a selection of political figures (section 5.1), to focus on the content of the hate messages (section 5.2). With this information, we select the most attacked and most violent politicians in the sample to perform a qualitative evaluation of the messages characterized as hate-speech (sections 5.3 and 5.4). Finally, in section 5.5. we infer the political orientation of users to establish a correlation between the political figures and their potential followers as the link between the two groups. We conclude the article with some conclusion at section 6.

## 2. Theoretical framework

This work analyzes the gender bias of hate speech in the political context of Argentina. Gender bias is fueled by sexism: the discrimination or prejudice based on a persons' gender. It is based on the belief that one sex or gender (male) is superior to another. It can take several forms, and one of these is sexist hate speech: a message of inferiority directed against women because they

are women, which can have an impact on the persons' physical integrity and inclusion [14]. Critical race theory and feminist theory also refer to these kinds of speeches as 'words that wound' [15].

Online communicative practices and, therefore, online vitriol, have a different form than offline ones. However, pre-existing cultural norms play a constitutive role in the way that 'hating' gets expressed online [16]. These 'words that wound' serve to marginalize and oppress those least privileged historically and in offline spaces—including women, people of color, trans people, and lesbian, gay, bisexual, and transgender (LGBT) people, and they have their counter-face as online hate speech.

Another aspect to consider regarding the dangers of hate speech is the harm it inflicts on democracies and the public sphere [17]. In real democracies many individuals are excluded from this public sphere by their race, sexual orientation, or gender [18]. New technologies restructured the public digital sphere [19] into a more flexible, fast, horizontal, and decentralized space, where the original inequalities of the public sphere persist and even worsen. The reproduction of these marginalizing practices inhibits oppressed collectives, making it inconceivable for them to develop their social life in a safe way, protected by the legal system [20].

Lastly, hate speech appears as a key element to manifest the 'politics of fear' that characterize the rise of right-wing populist governments [21]. Hate speech enables right-wing political parties and leaders to gain visibility and influence the public discourse, particularly on social networks where polarizing views often wield greater influence [22]. A striking example of this phenomenon is seen in the case of Jair Bolsonaro in Brazil. The former president employed hate speech through digital media, as evident from text mining analysis on his Facebook fanpage, which consistently featured expressions of partisan political animosity, as well as misogyny, LGBT-phobia, and xenophobia [23].

## 3. Related work

Thanks to the advancements in natural language processing tools, numerous studies have focused on automating the identification of hate speech on social media platforms [24–28]. While most studies have treated this task as a binary classification problem [24–30], efforts have been recently directed to pick out the different nuances of hate speech (e.g. the targeted group, whether the message contains a stereotype) and also to situate potentially hateful discourse within context [31–33]. These methodological advances enable us to consider fresh perspectives in studying hate speech on social media platforms targeting politicians and activists. For instance, Blanco-Alfonso et. al. [34] examined the potential gender biases present in violent mentions towards Spanish politicians and explored the most common derogatory terms used. The results of this analysis revealed that while Twitter messages directed at politicians exhibit a high level of hatred, there isn't a significant imbalance solely on gender. Nevertheless, a significant manifestation of sexism and misogyny was observed in the nature of attacks against women.

[35] identified the significance of specific users within social media in propagating antirights discourses through an analysis of Twitter data. Ozlap et. al. [36] employed machine learning tools to detect anti-Semitic speech and comprehend its dissemination patterns using a Twitter dataset. Their findings emphasize that counter-narratives to hate messages tend to spread more extensively and endure longer than hateful messages.

Twitter has proved to be a useful tool for political analysis. As it provides information regarding public opinion from users, it allows us to identify communities of interest and identify politicians occupying specific positions that are important for the dissemination of information [37]. Calvo & Aruguete [38] describe how Twitter appears as an "echo chamber" where

users see content that align with their previous beliefs. In the Argentine context, these previous beliefs can be polarizing, such as the case of trends related with the killing of Santiago Maldonado (#Maldonado) or some price rises (#Tarifazo), but might also depolarize the political scene, as in the case of the discussion for abortion rights (#AbortoLegal). The shapes of violence that appear exacerbated on Twitter are not only the result of what happens in this social media, but also the representation of a zeitgeist.

Even though Twitter is commonly used to analyze this kind of message, additional sources are also used in some studies. Vergani et al. [39] discovered that conspiratorial speeches targeting specific minorities during the Coronavirus pandemic originated from online hate speech against those identities, which varied according to the social and political context. To gather data, they examined a Telegram channel. Similarly, Rieger et al. [40] examined right-wing communities on Reddit to identify hate speech and its targets through topic modeling. They observed that these discourses were present in all the communities they analyzed.

## 4. Data & methods

### 4.1. Dataset construction

This paper uses tweets related to Argentine political figures as a source of information to study their connection with hate speech as both recipients and senders. The selection of political figures aimed to represent the main political parties in the Argentine electoral context towards the beginning of 2023. Four electoral blocks span the political spectrum. Firstly, Peronism/ Kirchnerism ("Frente de Todos" or FDT) and Radicalism/PRO/Macrism ("Juntos por el Cambio" or JxC) formed, at the moment of writing this paper, the majoritarian electoral alliances that have alternated in power over the last two decades. While both alliances could be identified closer to the political center of the spectrum, there are strong programmatic differences in political, economic and social terms. The FDT, rooted in Peronist tradition and progressivism, tends to express discursive positions in favor of state intervention in the economy, income redistribution policies, and civil rights, such as LGBT+ rights and the legalization of abortion.. In contrast, the JxC coalition tends to promote free-market politics and conservative stances in political and cultural terms. As a novel phenomenon in the 2023 electoral landscape, a radical right-wing electoral alliance called "La Libertad Avanza" (LLA) emerged, notably opposed to the feminist agenda, with a negationist approach regarding the genocide perpetrated by the last military coup in Argentina, which reflects on the Trumpist and Bolosonarist movements of US and Brazil. On the other hand, the left is represented by two blocks: the "Frente de Izquierda," with parliamentary representation, and the "Nuevo MAS".

At the time of defining the sample of political figures the candidates for the 2023 presidential elections were not defined. Therefore, priority was given to figures that met any of the following criteria: 1) had led electoral lists in previous elections, 2) held at any point a position in the executive or legislative branches or 3) represented a party within their respective electoral blocks. At the same time, party affiliations shown in this paper correspond to the moment of data capture but tend to be dynamic. For instance, Carlos Maslatón left LLA and has no longer explicit affiliation, while José Luis Espert moved from LLA to JxC.

As a result, 30 Argentine political figures were selected and all tweets (sampled via Twitter's API) mentioning them between February 2 and 9, 2023, were downloaded. This period was defined based on Twitter's announcement of discontinuing the Application Programming Interface (API) that allows information extraction [41].

Concurrently, tweets sent by these same political figures were retrieved. Due to API limitations, the most recent 3,200 tweets from each figure were collected, starting from February 9. Retweets were excluded to focus the analysis on discourse directly emitted by the politicians.

Nevertheless, the API includes retweets in the quota. Therefore, the collected information for each political referent varies in terms of tweet count and temporal span. Additionally, figures with fewer than 500 original tweets at the time of download were filtered out. This was the case for Javier Milei, who, although was a national legislator and presidential candidate for the LLA block, had fewer than 100 original tweets available for analysis due to the API limit being reached by his high number of retweets. Milei, however, was included in the analysis of hate speech reception.

The final dataset comprises 258,779 tweets mentioning the political figures and 44,430 tweets sent by the figures themselves. Fig 1A and 1B (later in the text) show the distribution of tweets sent and received by each political figure.

## 4.2. Hate speech classification methods

For each retrieved tweet, we analyzed the hate speech content using pysentimiento [13], a Python toolkit providing advanced classification algorithms in opinion mining tasks including sentiment analysis, emotion detection, and hate speech detection. Powered by pre-trained language models for several languages [42–44] and the Transformer architecture [45], the models are executed and distributed through the huggingface library [46].

We used one of the two hate speech classification algorithms available within the library. This particular algorithm was trained on a multi-label, finely-grained hate speech definition. Despite the original model incorporating contextual information for enhanced performance [32], we opted for this algorithm due to its training on Rioplatense Spanish data–aligning with the dialect of the collected data. In contrast, the alternative algorithm was trained on the HatEval dataset [24], largely composed of Peninsular Spanish text.

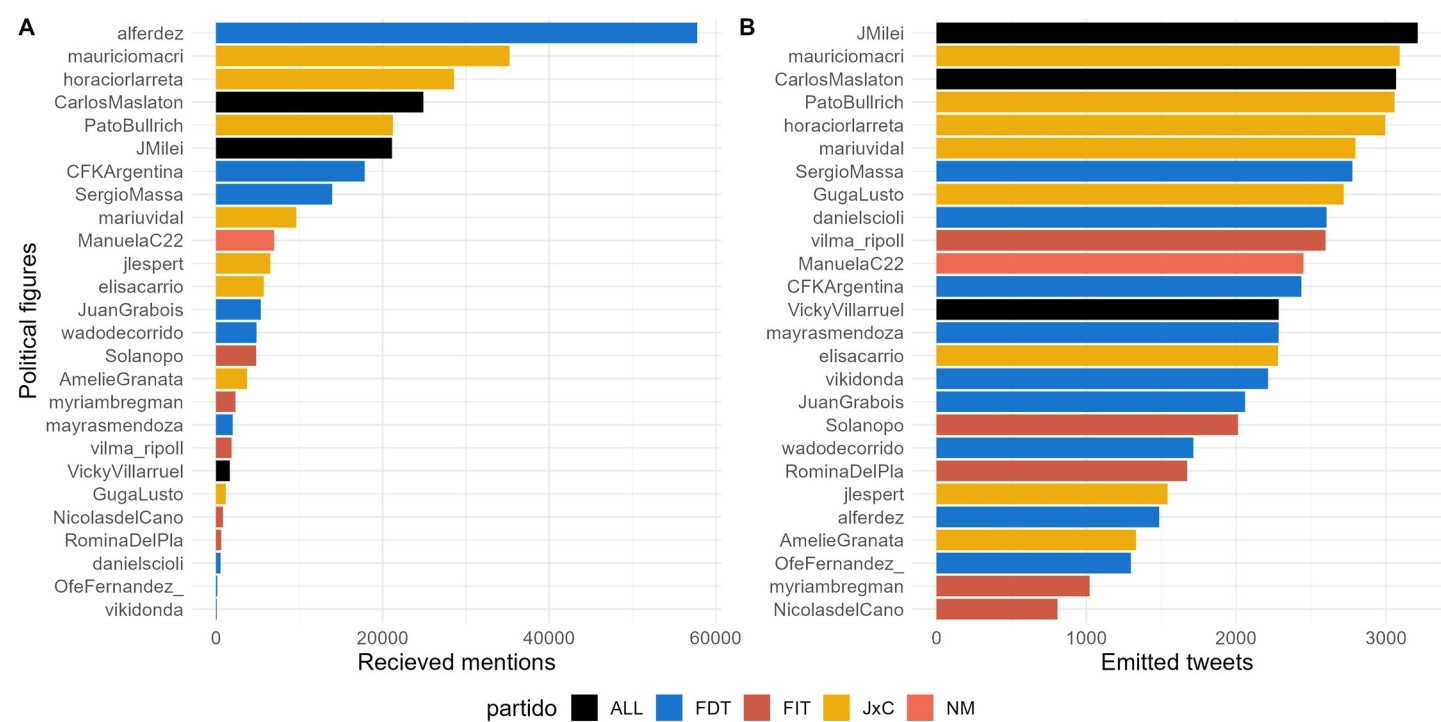

**Fig 1. Emission and reception of tweets. a**: Number of mentions directed towards political leaders between 02/02/2023 and 02/09/2023. **b**: Number of tweets issued by political leaders between 01/01/2020 and 02/09/2023, excluding retweets, bounded by the 5000 quota of twitter API.

This model yields a prediction of hate speech scores between 0 and 1 for eight distinct characteristics of hateful content (targeting women, LGBTQ+, racism, disability, criminality, politics, appearance), and also a score indicating the presence of a call to action. The model returns normalized scores, with values concentrated on the extremes, so a tweet was classified as hateful if any of the aforementioned categories scored above 0.5.

## 4.3. Detecting user political orientation

To have a better understanding of the directionality of hate speech, we inferred the political orientation of the commenting users. While detecting the political preferences could be performed by analyzing the graph homophily and the content produced by the users [47], we used a more straightforward approach based on user descriptions (also known as bios). Twitter bios can be treated as a proxy for an individual's sense of identity. When users add cues to their political affiliation there, it can be concluded that such affiliation is tightly tied to the users' identity [48]. Of course, this is limited to only those users that explicitly state their political orientation in their profiles. But nevertheless, results are representative for the most politically compromised users.

We constructed a dataset with weak labels using regular expression rules, leveraging the Argilla platform [49]. Each class was associated with rules established through prevalent expressions (e.g., "anti-k," "peronista," "zurdo") and references to user handles within bios (e.g., political figures, official party accounts). A selection of the top-matching regular expression rules can be found in S3 Table. This distant supervision process yielded 13,149 instances.

By observing users' description, we categorized political orientations into six classes: peronist, left-wing, macrist, libertarian, other-conservative, and neutral. Of these, three classes directly correspond to specific political parties or coalitions (peronist, macrist, libertarian), while another include a spectrum of different parties (left-wing). The remaining category accommodates profiles with conservative inclinations lacking explicit affiliations (other-conservative). This category includes pro-life, sexist, ultra-religious, nationalist, and other right-wing identities that characterize local conservative groups.

We trained a classifier for the multi-cassifcation task using RoBERTuito [44] as the pre-trained language model as it is specially tailored for Twitter data. Following standard practices [43], the classifier was trained for three epochs using a triangular learning rate and AdamW as optimizer [50] with a weight decay of 0.1, and $\beta_1$ and $\beta_2$ set to 0.9 and 0.999 respectively. Our model was trained using a GTX 1080 Ti GPU. To evaluate its performance, we labeled a separate set of 1,113 user descriptions manually. We can observe that a considerable number of errors are bounded to right-wing orientations, making them less harmful to our analysis (see S4 Table and S3 Fig).

## 4.4. Limitations

It is important to highlight some limitations and drawbacks of the proposed methodology. The first limitation is related to the socio-demographic bias in the analyzed social media. According to the National Survey of Cultural Consumption in 2017 (ENCC), less than 11% of the Argentine population used Twitter. These users were concentrated in certain profiles: high and upper-middle socioeconomic levels, young ages (from 12 to 29 years old), and in medium to large cities [51]. Recently, updated data from the ENCC for the year 2022 have been published, and the use of Twitter appears to have increased to 16% of the Argentine population [52]. At the time of writing this paper only preliminary results with aggregated figures of the 2022 survey had been published. This limitation implies that the scope of our analysis is

limited to the population of twitter users, which is nevertheless considered an important arena for the political discussion ongoing in the country.

There are, in addition, a set of limitations linked to certain challenges of the models used (and text classification models in general) in detecting certain aspects and nuances of natural language: it is well established that NLP models encounter difficulties in identifying irony, ambiguous words, slang, among other uses of informal language. These issues are particularly significant in the broader political discourse and within social media networks specifically. The use of hate speech detection algorithms for automatic content moderation raises some ethical concerns regarding the potential censorship of messages misclassified as hateful, and because it imposes a definition of hate speech that might not be shared across users [53].

To address and control some of these limitations, validation was performed through a more focused reading on a sample of tweets classified as hateful. In this way, the models employed serve as an initial filter that is subsequently validated through in-depth reading.

In the dataset used, some of these problems were detected. For example, the term "mariquita" has polysemy, and one of the meanings can be translated as "sissy", a hateful connotation that is captured by our models. But it can also function as a woman's name in Spanish. This is the case of Carlos Maslatón's (one of our political figures) wife. Consequently, many tweets sent or received by Maslatón were classified as hateful but were referred to his wife—in a non-hateful way—.

Given these limitations and the complexity of the task, we attempted to detect the directionality of the messages classified as hate speech: is it possible to detect whether a comment made by a given user and classified as hateful by pysentimiento is directed at the politician or not? Is the politician in question being targeted in that comment? To solve this task, we collected all tweets received by politicians that were classified as hateful by pysentimiento (15,187 tweets) and used GPT to classify each of them as either an "attack" or "non-attack" on the mentioned politician. We used GPT-4o-mini with a few-shot learning scheme in which the entire conversation thread and replies to each hateful comment directed at the political leaders were reconstructed. Then, the model was asked to classify the last tweet in the thread as an attack or not on the political figure. The prompt included, as contextual variables, a brief work and political description of the politician. The prompt text was:

"Decide whether the last tweet in the message thread represents an attack on the mentioned user. This attack may be direct and aimed at the person, or indirect towards their political space or identity (for example, towards feminism or their political group in general). For each tweet thread, we provide the username of the politician to be analyzed. Respond with 'attack' if the last tweet attacks the politician, and 'no attack' if it does not. Explain the reason for the decision in the 'Explanation' field and then answer whether the tweet attacks the politician or not in 'Output.'

Consider the following description of the user {description}."

As a validation test, we manually classified 100 comments. The manual classification was done with 3 annotators, achieving an agreement rate of 89.7% measured by Krippendorff's alpha. Then, GPT classified the tweets and obtained an F1 score of 89%.

## 5. Results

Not all political figures in our dataset belong to the same tier of influence. Therefore, our analysis starts by a contextualization of the number of tweets received and produced by each political leader. Fig 1A depicts the most mentioned political leaders in tweets, with the top positions occupied by key figures from the major parties in Argentina (FdT and JxC): Leading the list is Alberto Fernandez, the current president from FdT, followed by Mauricio Macri, the former

president from JxC. Subsequently, Horacio Larreta and Patricia Bullrich, the two presidential candidates for JxC, and Javier Milei, the presidential candidate for LLA, complete the top five positions. It's noteworthy that Carlos Maslaton (ALL), though not a candidate nor holding significant positions, gains attention for his active presence on social media, as shown in Fig 1B, where he ranks among the candidates publishing the most tweets.

## 5.1. Measuring hate speech

To investigate the relationship between the uptake and dissemination of hate speech by political figures, a two-fold corpus was compiled: tweets mentioning political leaders and tweets composed by the political leaders themselves. A tweet mentioning a user might function as a reply to a previous tweet from the politicians or exist as an independent message. The identified hate speech could either target the politician with an attack or express support for a violent message previously posted by the political leader.

Fig 2 illustrates the extent of hate speech emitted and received by political leaders, by showing the proportion of hate speech in both corpora. It can be observed that on average, and considering the number of tweets of each candidate, almost 8 out of every 1000 tweets issued by the candidates contain hate speech, while the average proportion of hate speech tweets mentioning candidates rise to more than 68 out of every 1000 tweets. This means that candidates receive eight times more hate than they produce. This discrepancy is expected, given that a significant portion of digital violence stems from the anonymity prevalent among many social media users [54], which political leaders do not possess.

There is a consistent observation that female politicians are more prone to receiving online hate speech compared to their male counterparts. Four out of five politicians that receive the most hate speech are women, and eight out of the thirteen that receive hateful messages above

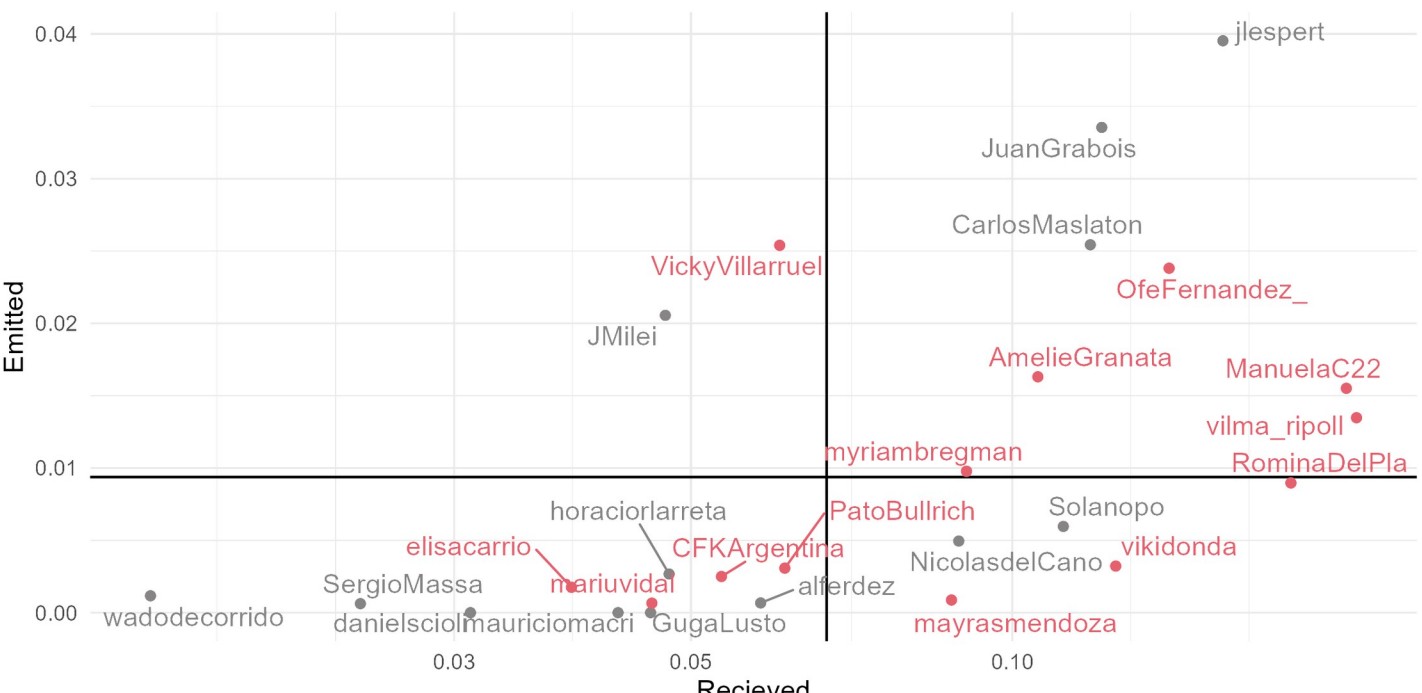

**Fig 2. Proportion of hateful tweets received and emitted by political figures, and weighted average by total number of tweets of each political figure.**

average are women, while only five out of the twelve politicians that receive less than average hate speech are women.

Concerning the dissemination of hate speech by political leaders, Espert, Grabois, Villarruel, Maslaton and Ofelia Fernández are the five politicians that stand out as the main proponents of hate speech. Of those, only Villaruel receives less hateful mentions than average. It is worth mentioning that Milei, the actual president who won in formula with Villarruel, is the only other politician that emits hate speech above average but receives it below average.

Nevertheless, the automatic detection of hate speech as such lacks the needed content and context of digital violence needed to fully understand the phenomena. For this reason, the next section will delve into a hate speech classifier that predicts the content of the violent discourse. We will also perform a qualitative analysis, informed by the automatic detection, to be able to understand the types of dialogues formed between politicians and other users.

## 5.2. The content of online digital violence

Hate speech is a complex concept that gathers many forms of discrimination. We have already seen that women politicians are a relatively more frequent target of hate speech (see Fig 2). Nevertheless, it is important to delve into the specific content that builds hate speech towards men and women politicians. To achieve this, we used a model that labels messages with eight possible offended groups plus whether the message contains a call to violent action (see methods above).

Fig 3 shows the proportion of hate speech on each of the specific categories addressed to political figures, divided by gender. The figure shows that women have higher values on almost all the hate speech categories, except for appearance, criminalization, and LGBTI topics. As expected on a corpus of tweets that mentions political figures, the category 'politics' is the largest on average, although insults referring to appearance—such as weight and age—have the

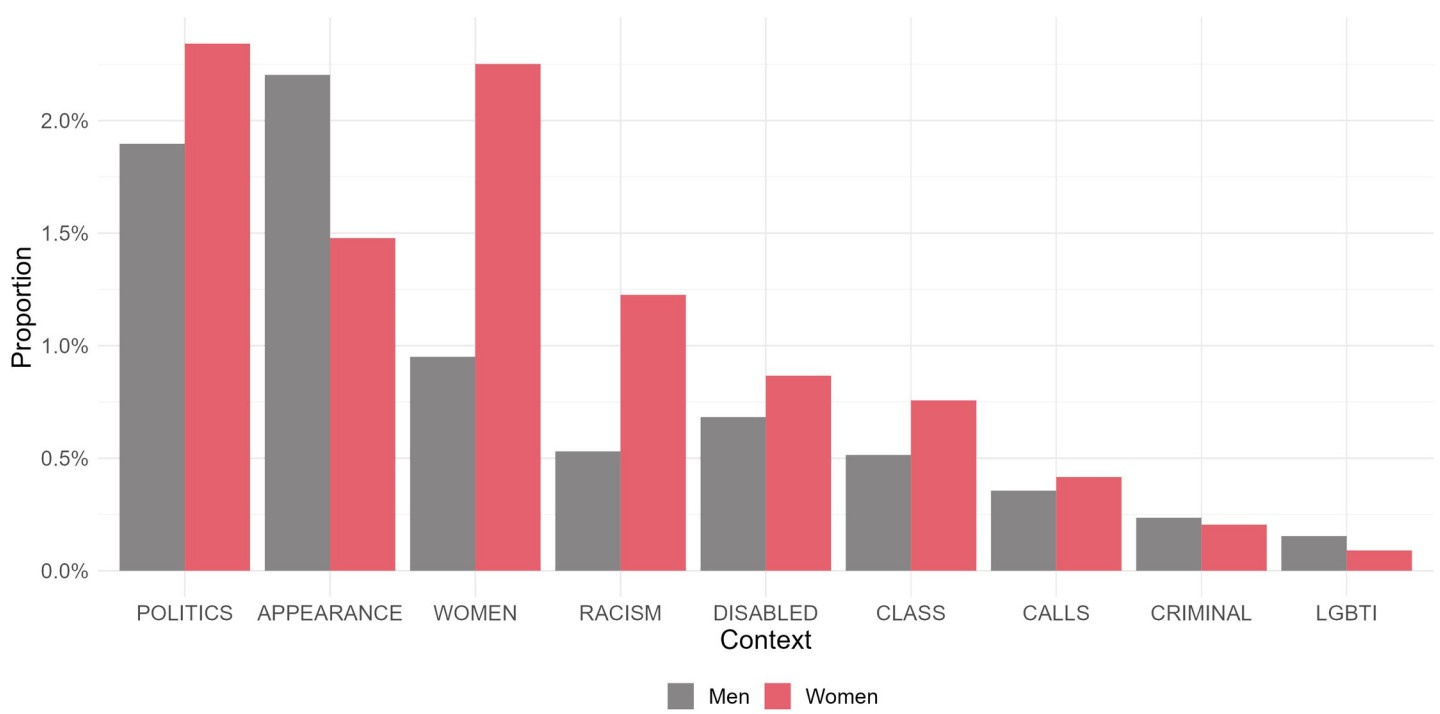

**Fig 3. Proportion of tweets that contain a specific type of hate speech over the total of mentions, by gender.**

largest proportion for men. The second largest group for women politicians is the category 'women', which englobes sexist comments.

### 5.3. Most attacked politicians

Large-scale automatic detection is useful as it helps us understand the big picture of the problem. Nevertheless, these models are also limited as they do not allow us to grasp the subtleties of the discourse produced. For example, a detected hateful message can be a direct attack to the mentioned politician, as well as an expression of support towards the hateful message that the mentioned politician proposes. To deep dive into this, we developed a qualitative analysis informed by the models. We selected the five politicians that received the largest proportion of hate speech and studied the most frequent terms used on those tweets that mentioned them and were classified as hate speech. It is worth mentioning that this group of the 'most hated' politicians—Castiñera, Ofelia Fernández, Del Pla, Espert, and Ripoll—is composed of 4 left or center-left women, and one right-wing man (Espert).

Table 1 shows the most frequent terms in the tweets that contain hate speech addressed to the 'most hated' political figures. The representatives of left-wing parties (Ripoll, Del Pla and Castañeira) are the only ones who receive the insults regarding their political ideology ('leftist'. 'communist'). In Ripoll's case, a figure of the FIT, we can also see insults related to laziness: the word 'lazy' itself and the combination of 'go' and 'work', which comes from a typical right-wing accusation in Argentina: 'vayan a laburar' (go to work). Romina Del Pla, another figure from the left-wing FIT, receives the term 'bullet' which is a common call towards violence, and the term 'lazy' again. Manuela Castañeira received nationalist-xenophobic content ('get out', 'country') and references to the country Peru ('peru', 'peruvian'). This is likely to be related with a statement against the military coup in Peru made by Castañeira on the week of the retrieval, which received a lot of hateful replies.

Messages towards Ofelia Fernandez contain many fatphobic insults ('fat', 'eat') and accusations ('murderer') that sometimes take conspiracy overtones ('cloned'). A worrying hashtag #hacerlamierda ('screw her up') which is an open call a violent physical attack. However, we also see the hashtag #colectivodeactrices which is a feminist actress collective.

In the case of the right-wing men politician, the most frequent terms seem to tell a different story. The words 'bullet' and 'jail' are among the most frequent, but they are also the most frequent terms used by Espert on the tweets that are classified as hate speech produced by the candidate (see Table 2). His name also appears within the most frequent terms. This points in the direction that the hate speech received by this politician is an expression of support for his

**Table 1. Most frequent words, excluding mentions, that appear on tweets classified as hate speech mentioning candidates.**

| RominaDelPla | vilma_ripoll | ManuelaC22 | OfeFernandez_ | jlespert |
|---|---|---|---|---|
| zurdo / leftist (masculine) | periodismodeizq / periodismodeizq | largat / get out | clonada / cloned | bala / bullet |
| zurda / leftist (femenin) | zurdo / leftist (masculine) | pai / country | morfa / eat | pelado / bald |
| vago / lazy | vago / lazy | sheraton / sheraton | hermana / sister | carcel / jail |
| anda / go | vilma / vilma | peru / peru | piensa / thinks | narco / narc |
| bala / bullet | vayan / go | peruano / peruvian | asesina / murderer | espert / espert |
| espert / espert | laburar / work | comunista / communist | foto / picture | vago / lazy |
| jxc / JxC | zurda / leftist (femenin) | | gorda / fat | |
| | enclav / enclave | | ve / see | |
| | chino / chinese | | bolsa / bag | |
| | | | colectivodeactric(hashtag) | |
| | | | hacerlamierda (hashtag) | |

**Table 2. Most frequent words, excluding mentions, that appear on tweets classified as hate speech produced by candidates.**

| CarlosMaslaton | JuanGrabois | OfeFernandez_ | VickyVillarruel | jlespert |
|---|---|---|---|---|
| loko / crazy | gorila / gorilla | sujeta / held to | zurdo / lefty | carcel / jail |
| judio / jewish | grave / serious | proyecto / project | mapuch / mapuches | bala / bullet |
| bullish / bullish | pobr / poor | va / goes | patagonia / patagonia | sitio / place |
| israel / israel | pueblo / people | victima / victim | ano / year | zona / zone |
| pai / country | planero / planero | cambiando / changing | delincuent / criminal | delincuent / criminal |
| argentina / argentina | violento / violent | caso / case | gobierno / government | simio / monkey |
| gracia / fun | amigo / friend | derecho / right | mujer / woman | villa / slum |
| alcohol / alcohol | ano / year | desalojo / eviction | tierra / land | justicia / justice |
| comunista / communist | cristina / cristina | emergencia / emergency | argentino / argentine | terrorista / terrorist |
| consumo / consumption | justicia / justice | feminista / feminist | canich / poodle | vida / life |
| frasco / jar | lei / read | mierda / shit | feminismo / feminism | |
| | medio / means | mundo / world | lei / read | |
| | negro / black | policia / police | millon / million | |
| | odio / hate | respond / answer | politica / politics | |
| | politico / politic | tran / transgender | quieren / want | |
| | preso / imprisioned | tratar / deal with | xq / why | |
| | social / social | travesti / transvestie | | |
| | vulner / vulnerable | vive / lives | | |
| | | voto / vote | | |

own hate speech rather than an attack on his person. Interestingly, his name is also one of the most frequent terms for Del Pla, which gives a hint on the directionality of the political aggressions that flow in this social media. Finally, we do see terms like 'bald' which are probably aggressions related with the alopecia of the candidate.

To validate the above analysis, we selected a random sample of 15 tweets from the 'most hated' political figures that were automatically classified as hate speech (see S1 Table). In this way, the language models inform the qualitative analysis by pointing us to the needed sample of cases.

Our first finding was that many hateful tweets against the female candidates came from replies towards something they tweeted regarding a political or social situation. Much of the hateful tweets received by Castañeira are a response to her positioning against repression of the military coup in Peru. As a foreigner to Peru, there are many xenophobic reactions, intertwined with ideological and gender hate—'go back to your country feminazi'; 'fucking Argentinean, go back to your country, fix it, and then come back'; 'got back to your country, burned by socialism, leftist failiure',etc—. The qualitative analysis of those responses also allows us to see that many of those uses are from Peru and not from Argentina. In the case of Del Pla, this happened several times as responses to her tweets regarding her bill projects: '@DiputadosAR bullet for you too'; @FdeIzquierda @DiputadosAR go to work (racist comment against black people and lesbians)'. We can also see that these comments come from Espert supporters—'@EBelliboni @RominaDelPla @jlespert #todosconEspert (all with Espert) 💪impoverishing lefties.', or are comments against Espert but not targeting Del Pla—'@RominaDelPla But what an aggressive and violent old man, that Espert.'—

For Ripoll, most were replies towards a tweet denouncing the repression of activists in the land struggles in the south of Argentina. These tweets are violent and targeted against Ripoll, they also mention the user Cele_Fierro, another female left-wing politician that she was mentioned in the original tweet and refer to her political ideology as an insult: 'run leftist';

*'@vilma_ripoll @Cele_Fierro You are also disgusting, with your green scarf,'* (the green scarf is the symbol of legal abortion campaign); *'brainless lefty'*. There are also comments associated with the "lazy," "foreigner" status of the protesters, which are classic topics in right-wing discourse.

For other political figures, violence is not a response to their actions, but a constant. This is the case of Ofelia Fernandez. For example, news about a policial case of a lesbian couple that killed a child was used as an excuse to attack Ofelia Fernandez as a feminist, adding references to her body image. Also, in our random sample of 15 cases we retrieved five tweets from the same user mentioning a conspiracy theory of a cloned sister of Fernandez, which reflects on a systematic type of violence, rather than spontaneous social media comments. This user repeated the same tweet several times (each time mentioning different users), which could indicate that it is a troll.

When we look at sampled tweets that mentioned Espert, we confirm that most of those tweets are not hate speech that targets him, but reinforcing what he says and a general call to violence. With many references to 'jail or bullet', these messages replicate Espert lemma. Two of the 15 tweets sampled seem to be in opposition to Espert: *'@jlespert You ask the poor for a bullet and for your drug dealer friend (nothing)?'; '@jlespert Deranged gorilla, your embittered existence surfaces everywhere, your skin sweats poison, what a pity it doesn't cause poisoning and you stop dirtying the planet.'*. Even if they have an aggressive tone, those messages against Espert are far less violent than those in support and in clear antagonism against the left: *'@jlespert Just kill them all'; '@jlespert Bullet* (they) *can't be fixed'*.

When we return to the model and compare the hate speech contextualized in different types (see S1 Fig), we see that the model is consistent with the qualitative analysis. Tweets that mention Espert have higher scores in calls to action. Tweets against Castañera rank high in racist speech; and together with Del Pla and Ripoll, they have high scores in political hate speech, while Ofelia Fernández ranks high in the category 'women'.

Given our findings through the qualitative analysis, we conducted complementary experiments to detect the directionality of the hate speech received by politicians. We want to understand if the messages classified as hateful are an attack on the mentioned politicians or not. To address this, we used GPT (see data & methods). Each hateful comment received by political leaders was classified as either an 'attack' or 'non-attack' against that politician. Then, we recalculated the proportion of hateful tweets received, weighting it by the proportion of tweets classified by GPT as 'attacks.'

Fig 4 shows the difference between the proportion of tweets for each leader before and after weighting by directionality. Logically, all leaders show a certain decrease in the proportion of hateful tweets received. However, it can be observed that some leaders, for whom we detected some classification issues in the original model, show a significant drop, while others only decreased marginally. Espert's proportion of hate speech decreases from 0.16 to 0.031, and Maslatón's from 0.12 to 0.068.

To provide an overview of the changes, Fig 5A shows the proportion of attacks according to the political orientation of the leaders. These results highlight how the hate speech related to right-wing candidates is much less related to direct attacks on those figures (with almost a 50% decrease). On the contrary, the hate speech detected for FDT and left-wing leaders was much more consistently a direct attack targeting those political leaders, with over 75% of the messages labeled as direct attacks.

Finally, Fig 5B shows that the gender of the political leader remains a relevant variable in the proportion of attacks: Women receive a higher proportion of attacks across all political parties, with the exception is FDT where the rate of attacks on women is slightly higher than that of men. Furthermore, gender strongly interacts with political orientation: women from

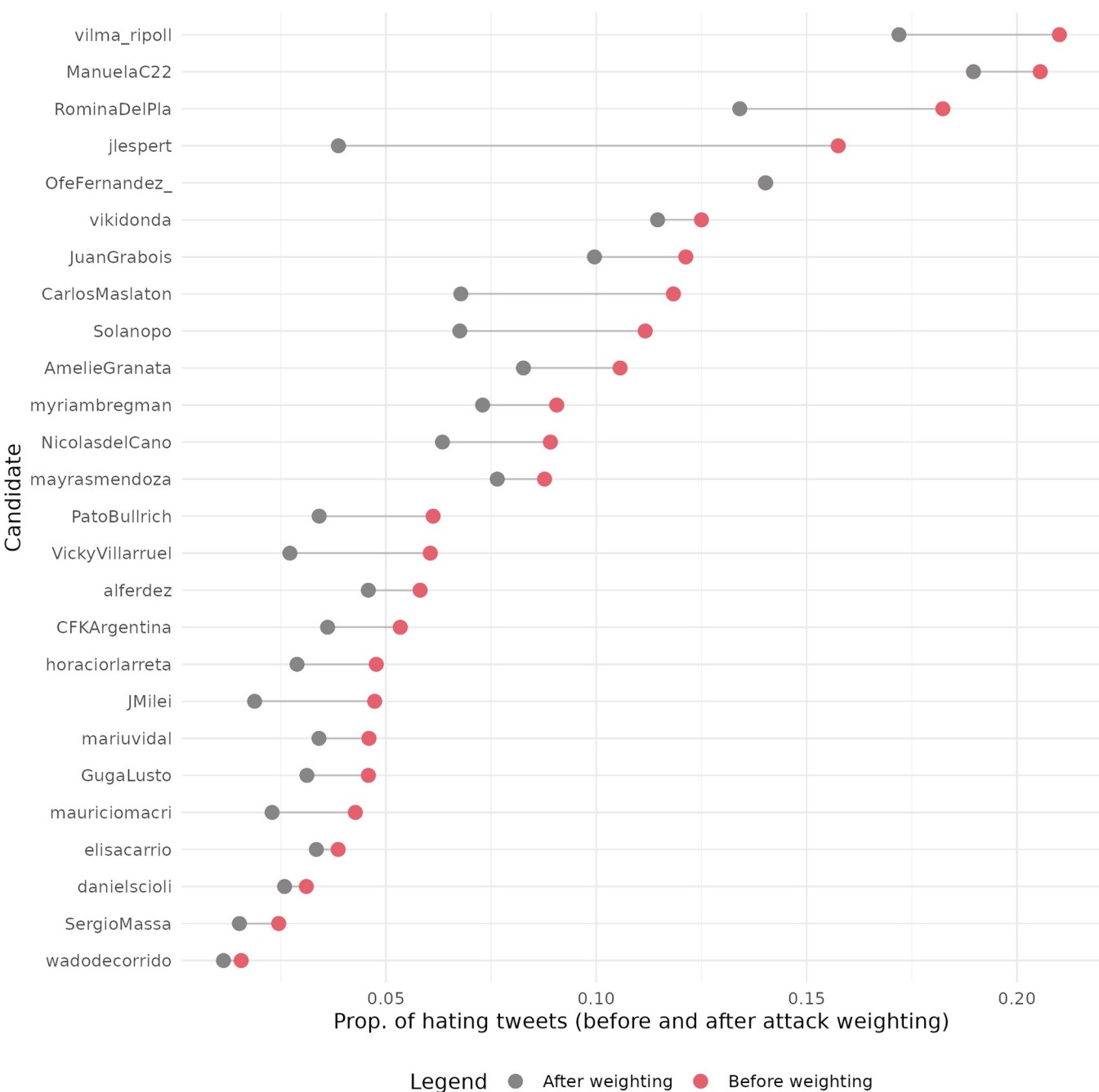

**Fig 4. Proportion of hate tweets received by politicians before and after weighting by directionality.**

FDT and the left receive 72% and 88% of attacks, respectively. This proportion decreases to 45% in the right wing, and 66% in JxC.

## 5.4. Most violent politicians

Analogously to the previous section, we studied the politicians that ranked highest in hate speech messages, with focus on the most frequent terms and a qualitative evaluation of a random sample of tweets that were predicted as hateful. This analysis will have a double role: first,

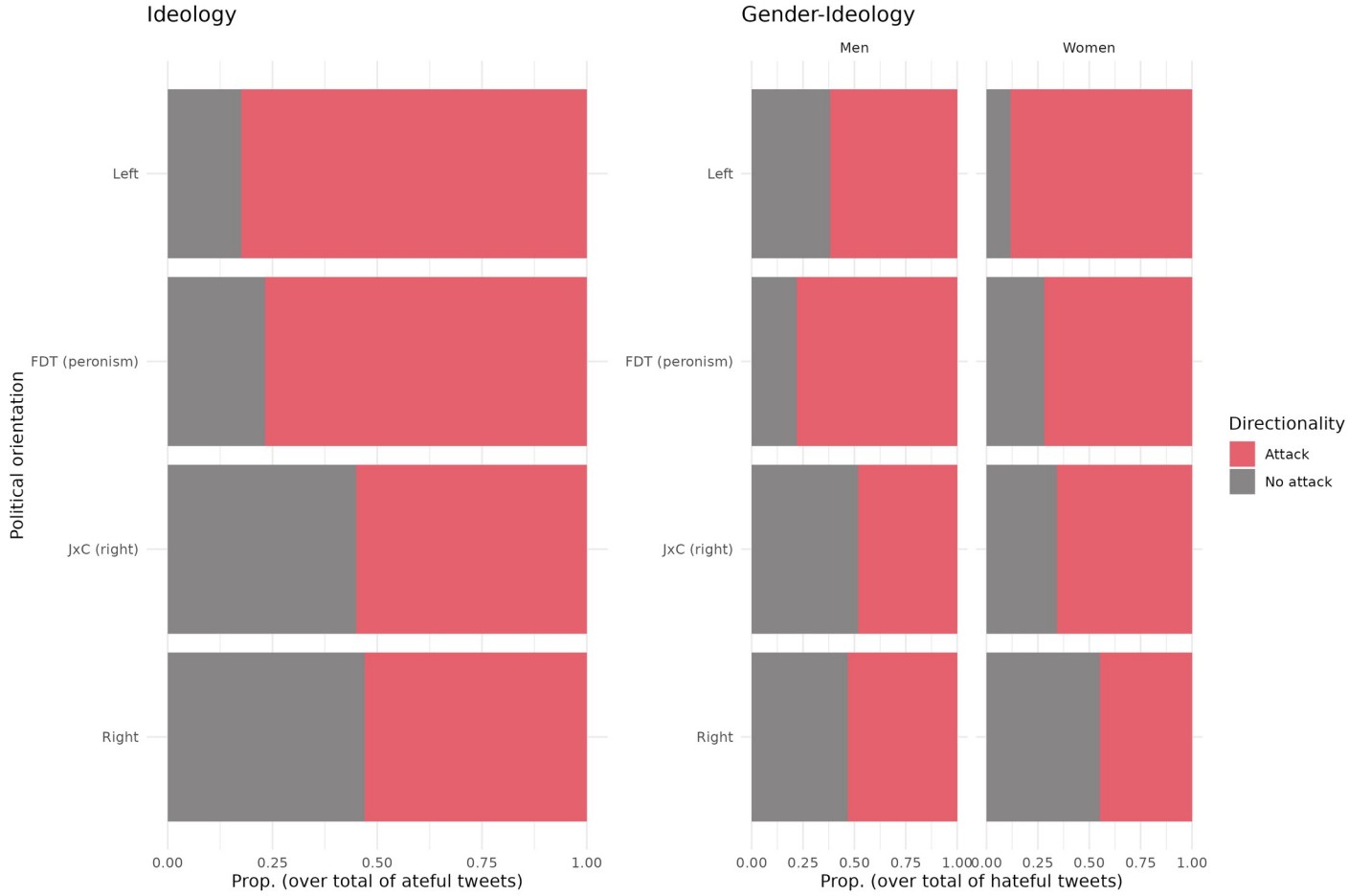

**Fig 5.** Proportion of hateful tweets emitted by politician's political orientations (A) and gender (B).

it is a validation of the model, to check whether hate speech is correctly detected. Second, it will allow us to understand the relation between the candidates' political orientation and how this relates with the hateful rhetoric they spread in social media. The politicians that rank highest in hate speech are Espert, Villarruel, Maslatón, Grabois and Ofelia Fernández. Contrary to what happens with those that received the most hateful tweets, this groups are composed by three men—Espert, Maslatón, and Grabois—and two women—Villarruel and Ofelia Fernández—, and three right-wing political figures—Espert, Villarruel, and Maslatón—and two center-left peronists—Grabois and Ofelia Fernández—.

Table 2 shows the most frequent words coming from automatically detected hate speech tweets. For most candidates there is at least one reference to minorities or collectives. For Maslatón, words referring to the jewish community appear ('jewish', 'israel'). Nevertheless, this politician is jewish and an advocate of Israel. As insults against this minority are an extended form of hate speech, the model is trained to give high scores to tweets that contain those words, and therefore these are probably false positives. On the other hand, his mentions of communist ('comunista') are probably of a derogatory character. In the case of Grabois, who is a political leader of the *piquetero* movement, unemployed lower-class people that receive state aid, many words refer to poverty ('poor', 'planero','vulnerable', 'social'). The term 'planero' is an interesting example. Normally used in a deceptive manner referring to people

that received state aid, Grabois—as a leader of that movement—is either reappropriating the concept, in a similar way the LGBTIQ+ community reclaimed the concept of queer [55], or discussing with people that utilize the term. His detected tweets also discuss the subject of hate speech itself ('violent', 'hate'). Nevertheless, the term 'gorilla' is a typical derogatory term from the peronist movement towards the right-wing. Ofelia Fernández, from the same political space as Grabois, there are references to the feminist and LGBTIQ+ movements ('transvestite', 'transgender', 'feminism'), which are probably false positives as she is a feminist activist. There are also terms related to politics ('project', 'right'), and the word 'shit'.

On the other hand, the word 'feminism' for Villaruel will probably have a different content than in the case of Ofelia Fernández, as she is openly opposed to that movement. This shows how in the case of politicians, there are a lot of subtleties that need to be considered for each specific case. Villaruel also makes references to the conflict of the Mapuche people, which are in a struggle for land ('mapuches', 'patagonia', 'land'). This is the same land struggle for which we detected several hateful messages towards Ripoll (see above), and that were mentioned in responses to Espert. This example shows how an ongoing debate that splits waters between different political positions appears as hate speech three different ways. First, as hateful messages that common users produce against left-wing candidates. Second, as hateful messages that common users produce in favor of a right-wing candidate. And finally, as hateful messages that emerge from the right-wing figures themselves. Espert is probably the case where there is not much space to doubt about false positives from the model. The word 'monkey' next to 'slums' is a typical racist collocation of terms against poor people. 'Criminal' and 'terrorist' might be associated with the Mapuche struggle in line with Villaruel's comments. Finally, 'jail' and 'bullet' are open calls to action.

The above analysis sheds some light over the content of hateful tweets from candidates, but also shows the importances of a close reading of a sample of tweets to better understand the produced discourse and the eventual false positives from the model. For this, we randomly selected 15 tweets of each candidate (see S2 Table).

For Maslatón, we see a mix of messages with vocabulary from financial markets–*'bull market'*, *'bullish'*—, a lot of mentions to his relations with drugs, some informal slang with racist connotations—*'groncho'*, *'negro'*—but used to make positive references to those groups. Finally, we see aggressive uses of the term *'communist'* as a form of insult.

In the cases of Ofelia Fernández and Grabois, there is also a widespread use of slang and insults, but from our point of view these messages are not within the scope of hate speech. This can be seen in Ofelia Fernández's tweets: ' *(. . .) if you are going to yell die you son of a bitch you are a neighbor, if you are going to support Cristina you are an identified aggressive militant'*; ' *(. . .) What a piece of shit once justice does not screw up a policeman goes and violates the victim (. . .)'*. In other cases, the tweets address the issues faced by marginalized populations—*'The transvestite-trans population often wins per day and lives in hotels. In this context they have nothing and increase the price or want to evict them.'*—. This can also be seen in Grabois tweets where he condemns discrimination—' *'Jews are violent', 'Christians are violent', 'Homosexuals are violent', 'The Mapuches are violent' Discriminatory and illegal generalizations'*.

For other political leaders, the tweets automatically flagged as hate speech poses offensive content against minorities.

Villarruel's tweets make direct attacks on the feminist movement and left-wing ideologies —*'They don't care that millions don't have to eat as long as they can continue with their talks about feminism, abortion, inclusive language and other leftist things.'*—. Her tweets also make references to the last military coup, the land struggle in Patagonia and the Native communities —*'The leftist message is that if a woman is a criminal and uses her pregnancy or her children as a shield and excuse, she cannot be stopped. Just like in the 70s where the montoneras did the*

*same while committing serious crimes. Whoever commits a crime must be punished. #mapuches'*—The coup was for Argentina's history the consummation of hate speech in the torture, murder and disappearance of thirty thousand people accused of being left-wing activist.

In the case of Espert, many of his tweets classified as hate contain direct calls to violence. The phrase 'jail or bullet' is repeated as a lemma, associated with the Mapuche people, and the piquetero movement—*'Jail or bullet for these criminals.'; 'No dialogue with these Maputrucho terrorists who question Argentina's sovereignty in the area. State of siege first and then jail or bullet. And the human rights organizations that stop screwing around and once defend innocent citizens.'*.

Looking back at the hate speech by the model (see S2 Fig), we can see that the model detects Espert's calls for action, and Villarruel attacks against the left-wing. It also helps to understand the caveats of the model, as Grabois ranks high on 'class', because he is talking about the poor, and Ofelia Fernández ranks high in 'women', as she talks about the feminist movement. In this sense, the combination of the disaggregated model together with the proper contextualization of the politicians allows us to better understand the patterns of hate speech arising from the political leadership.

## 5.5. The political alignment of users

We've shown the increased likelihood of women politicians, particularly those on the left and center left, receiving hate speech on Twitter. Our analysis, both quantitative and qualitative, has also revealed that right-wing politicians, regardless of gender, tend to engage in promoting hate speech through their accounts more frequently. There is a missing gap in understanding the political orientation of those users that are not public figures, and how that correlates with the gender bias that was found. To address this aspect, we developed a model that predicts a user's broad political alignment based on their profile description (refer to data & methods). Despite the fact that most users do not explicitly mention their political orientation in their profiles, we selected a subset of users for whom we were able to make predictions in order to analyze the distribution of hate speech. It should be noted that this analysis does not aim to establish a causal connection between politicians' hate speech and that of other users, as it falls outside the scope of this paper. Instead, its purpose is to have a first approximation to analyze from which social groups hate speech comes from, and how the gender bias is built.

Fig 6 shows how tweets are distributed by political orientation of the sender (general users), and the gender of the receiver (the politicians). Fig 6A shows the total number of tweets, while 6b shows the proportion of hateful tweets by group, sorted by their average proportion of hate speech. As expected, the orientations that produce the higher number of tweets are Macrism (related to JxC) and Peronism (related to FdT), which represent the largest political representations. For all groups, male politicians receive more mentions than women, which is also seen in Fig 1. When we look at the proportion of hate speech produced by groups, we see that libertarians (related with ALL), other right-wing users (that cannot be associated with a specific right-wing faction) show the most hateful discourse. We can also see that the gender bias is not equally distributed across political orientations. Libertarians, macrist, and especially other right-wing users show the largest animosity against women politicians, while peronists do not show an imbalance, and left-wing users tend to aggravate more male politicians. This confirms the previous literature that stated that right-wing groups used hate speech as a tactic against women politicians.

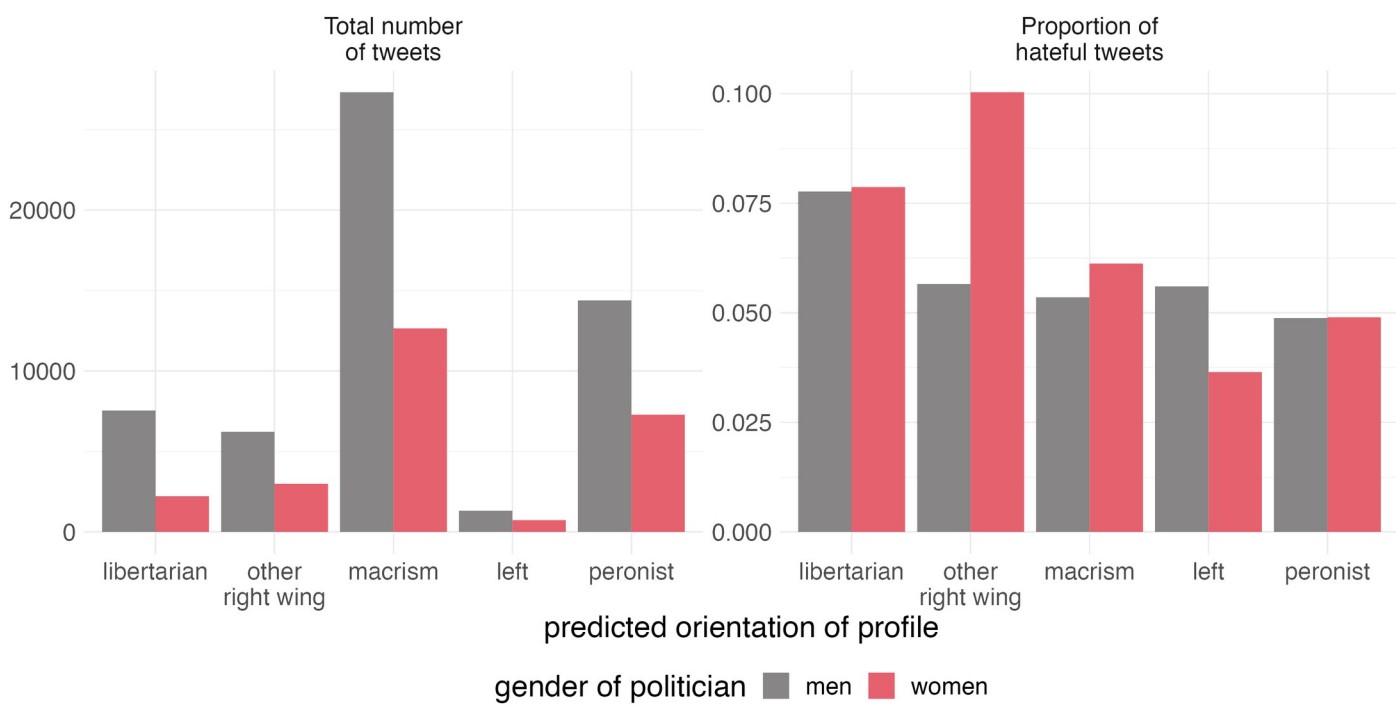

**Fig 6. Total number of tweets emitted by users' political orientations and proportion of hateful tweets by gender of politicians.**

## 6. Conclusion

Our large-scale analysis of tweets towards and from Argentine politicians, combined with deep learning models for automatic detection of hate speech and prediction of political orientation of users, together with a guided qualitative exploration of cases allows to shed new light over the flows of hate speech in social media. Several findings arised as contributions to the discussion of hate speech in Argentina and the region: First, that the distribution of mentions, with and without hate speech, has a gender imbalance. Even when our dataset is composed of 13 men and 13 women leaders across the political spectrum, men received more than 70% of all mentions. There is a correlation between the political power of these leaders and the number of interactions they received, but this does not nullify the fact that women politicians receive less impact on social media.

Nevertheless, this greater impact that men have on mentions does not prevent women from being the victims of most of the hate speech, receiving 57% of tweets flagged as hate speech. When we study with greater detail the hate speech produced against those politicians that receive most of the aggressive content, we noticed that left-wing women are the main target of political violence, and their political orientation is a main subject of such violence. On the contrary, right-wing candidates that receive large amounts of hate speech do so in the form of a validation to their own discourse.

Politicians tend to receive more hate speech than what they generate. Nevertheless, in our analysis we found that some political leaders promote hate speech through twitter. By studying the content of tweets flagged as hate speech, we observe that for small samples (26 politicians) the model raises false positives. The quantitative and qualitative analysis of the most salient cases allow us to better comprehend the discursive context in which some vocabulary associated with hate speech is used and disregard those cases. In other cases, such as Villarruel and Espert, right-wing leaders that belong to different political spaces, the automatic prediction of

hate speech was right, finding several examples that could be associated within the concept of hate speech. For these candidates, the main targets of hate speech are native communities, left-wing, feminist, and LGBTIQ+ movements. Also, open calls to action were repeatedly found.

Finally, when we look at the tweets that mention politicians from the perspective of the political orientation of users, we can see how those users that are aligned with right-wing politicians are the ones that reproduce and multiply that hate speech, with a particular bitterness against women.

Our automatic models suffer from some limitations. False positives arise when irony is used or when minorities are being mentioned in a non-hateful way. Nevertheless, this is the case especially for small samples, such as the 26 politicians from our database. Even with those limitations, these models can still be a useful guide to sample cases for a careful read, which in turn allow us to reinterpret the results of the model with an informed read. Taken together, our analysis brings new insights over a pressing issue for the argentinian society, given its long term history of military coups materializing violent discourses into state genocide, and the more recent attempt of magnicide perpetrated by extreme right-wing groups. Our work provides a novel mixed methods approach for detection and comprehension of political violence towards women on social media, that can be reproducible for other countries or regions.

## Supporting information

**S1 Fig. Proportion of hateful tweets received by political figures, by categories of hate-speech.**
(TIF)

**S2 Fig. Proportion of hateful tweets emitted by political figures, by categories of hate-speech.**
(TIF)

**S3 Fig. Confusion matrix of predicted labels of users' political orientation.**
(TIF)

**S1 Table. Sample of hateful tweets mentioning candidates.**
(XLSX)

**S2 Table. Sample of hateful tweets from candidates.**
(XLSX)

**S3 Table. A selection of the regular expression rules applied for weak labeling of users' profiles.**
(XLSX)

**S4 Table. Classification results of the RoBERTuito model trained for political orientation detection.**
(XLSX)

## Author Contributions

**Conceptualization:** Laia Domenech Burin, Juan Manuel Pérez, Germán Rosati, Magalí Rodrigues Pires, María Nanton, Diego Kozlowski.

**Data curation:** Laia Domenech Burin, Magalí Rodrigues Pires, María Nanton.

**Formal analysis:** Laia Domenech Burin, Diego Kozlowski.

**Investigation:** Laia Domenech Burin, Germán Rosati.

**Methodology:** Laia Domenech Burin, Juan Manuel Pérez, Germán Rosati.

**Software:** Juan Manuel Pérez.

**Supervision:** Laia Domenech Burin, Diego Kozlowski.

**Validation:** Laia Domenech Burin, Juan Manuel Pérez, Diego Kozlowski.

**Visualization:** Laia Domenech Burin, Diego Kozlowski.

**Writing – original draft:** Laia Domenech Burin, Juan Manuel Pérez, Germán Rosati, Diego Kozlowski.

**Writing – review & editing:** Laia Domenech Burin, Juan Manuel Pérez, Diego Kozlowski.

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
