## [Decision Letter · Decision Letter 0]

20 May 2024

PONE-D-24-03284Gender biases and hate speech: promoters and targets in the Argentinean political contextPLOS ONE

Dear Dr. Domenech Burin,

Thank you for submitting your manuscript to PLOS ONE. After careful consideration, we feel that it has merit but does not fully meet PLOS ONE’s publication criteria as it currently stands. Therefore, we invite you to submit a revised version of the manuscript that addresses the points raised during the review process.

We look forward to receiving your revised manuscript.

Kind regards,

Stefano Cresci

Academic Editor

PLOS ONE

Journal Requirements:

3. Please remove your figures from within your manuscript file, leaving only the individual TIFF/EPS image files, uploaded separately. These will be automatically included in the reviewers’ PDF.

4. Please upload a copy of Table S1 to S5 and Figures S1 to S6 to which you refer in your text. Please amend the file type to 'Supporting Information'. If the Supplementary file is no longer to be included as part of the submission please remove all reference to it within the text.

Additional Editor Comments:

The two reviewers made opposite recommendations. After considering all comments I opted for an R&R decision as I believe you have the possibility to address reviewer#2's concerns. However, I expect this to be a difficult revision. In case you think that you will be able to edit the manuscript and fix all outstanding issues, and particularly those of reviewer#2, please submit a revised version. As I see it, you will need to add a dedicated Related Works section and to discuss the contributions of this work with respect to a broader set of works, including state-of-the-art computer science ones. Then, reviewer#2 also suggested the use of more recent and powerful classification methods, such as LLMs. Additional experiments in this direction can reveal the extent to which the current classifications are accurate and reliable. Finally, reviewer#1 demanded a better theoretical grounding of the work. Please, also consider all remaining and possibly minor issues that I did not explicitly mention here.

Reviewers' comments:

Reviewer's Responses to Questions

**Comments to the Author**

1. Is the manuscript technically sound, and do the data support the conclusions?

Reviewer #1: Yes

Reviewer #2: No

2. Has the statistical analysis been performed appropriately and rigorously? 

Reviewer #1: N/A

Reviewer #2: Yes

3. Have the authors made all data underlying the findings in their manuscript fully available?

Reviewer #1: No

Reviewer #2: Yes

4. Is the manuscript presented in an intelligible fashion and written in standard English?

Reviewer #1: Yes

Reviewer #2: Yes

5. Review Comments to the Author

Reviewer #1: This work has been made available in https://osf.io/preprints/socarxiv/6cts8

---

The paper provides a comprehensive overview of hate speech in the context of social media, particularly focusing on its impact on marginalised groups and its connection to political agendas in Argentina. It effectively outlines the current state of research, methodologies, and notable findings in the field. However, there are areas where the paper could be strengthened to enhance clarity, depth, and academic rigor.

The paper effectively presents existing literature on hate speech, providing a solid foundation for the study. It covers various aspects, including definitions, historical context, technological advancements, and regional variations, demonstrating a comprehensive understanding of the subject matter. By incorporating studies from different regions and platforms, the paper offers a nuanced understanding of hate speech dynamics. It acknowledges the contextual nature of hate speech and highlights the importance of considering diverse social and political contexts in analyzing this phenomenon. The research questions are relevant and timely, addressing gender bias, political figure involvement, and replication of digital violence. These questions are crucial for understanding the mechanisms underlying hate speech dissemination and its impact on society.

While the paper provides a comprehensive review of existing literature, it lacks a clear theoretical framework to guide the analysis. Incorporating theoretical perspectives from communication studies, sociology, or critical discourse analysis could enhance the paper's theoretical depth and analytical rigor. Moreover, the paper briefly mentions methodological approaches such as text mining, machine learning, and topic modeling but lacks detailed descriptions of the methodologies employed in the studies reviewed. Providing more information about the specific methods used in hate speech detection and analysis would help readers evaluate the validity and reliability of the findings.

While the paper acknowledges the contextual nature of hate speech, it could benefit from a more in-depth exploration of regional variations, particularly in the Latin American context. Providing case studies or examples of hate speech dynamics in specific regions would enrich the analysis and facilitate a more nuanced understanding of the phenomenon.

Hate speech detection and analysis raise ethical concerns related to privacy, bias, and potential harm to vulnerable communities. The paper could benefit from a discussion of ethical considerations inherent in hate speech research and how researchers address these concerns in their methodologies and practices.

Reviewer #2: The study examines gender biases and hate speech in Argentine politics by analyzing the correlation between political affiliations and the propagation of hate speech, particularly against women politicians. The main problem with this work is that the authors do not provide a clear indication of how this research is positioned within the context of existing state-of-the-art works, making it very difficult to evaluate its scientific contributions. Another potential limitation is that the machine learning models used in the study struggle with detecting nuances in natural language, such as irony, slang, or ambiguous expressions. This limitation could be significant given the context of political discourse on social media, where such nuances are common and critical for accurate analysis. Moreover, the findings may not be easily generalizable beyond the specific social media environment and demographic profile of the users studied, limiting the applicability of the conclusions to other platforms or broader demographic groups.

In the following, some suggestions to improve the quality of the work:

- A “Related work” section is completely missing and without it it’s very difficult to compare this work with state of the art methods and consequently the impact of your scientific contributions. Please integrate such section in order to report which results and findings have been obtained in other existing works and how eventually your work differs from existing ones.

- The use of advanced commercial or open-source large language models (LLMs) (e.g., ChatGPT, Mixtral, etc.) as zero-shot classifiers for automatic classification of aspects of interest (sentiment, hate speech, etc.) should be considered in your work as a better alternative to overcome some of the limitations of the ad-hoc models used in this study. In particular, such LLMs are generally very capable of analyzing text correctly and contextualizing input text, even in cases where content originates from social media (e.g., slang, emojis, etc.). In addition, such tools are able to properly interpret a text by determining if some hate speech is against someone specific, not just by considering if in the text there are specific keywords or user mentions that coud confuse a basic NLP method based on traditional machine learning. Therefore, the use of such tools could provide more accurate results, ultimately improving the overall quality of your findings.

- To enhance content organization and clarity, I recommend dedicating a separate section to discussing potential limitations of the methodology applied in this study. This approach will enable readers to focus more easily on the research design and methods used, while also reflecting on aspects related to the generalizability of the findings.

- Unable to find in the article some referenced tables and/or figures (e.g. Table S5, Figure S6, etc.).

6. PLOS authors have the option to publish the peer review history of their article (what does this mean?). If published, this will include your full peer review and any attached files.

Reviewer #1: **Yes: **Marjory Da Costa Abreu

Reviewer #2: No

---

## [Author Response · Author response to Decision Letter 0]

2 Nov 2024

Journal Requirements:

Answer:

We have modified the format to adapt it to PLOS ONE style.

Please note that PLOS ONE has specific guidelines on code sharing for submissions in which author-generated code underpins the findings in the manuscript. In these cases, all author-generated code must be made available without restrictions upon publication of the work. Please review our guidelines at https://journals.plos.org/plosone/s/materials-and-software-sharing#loc-sharing-code and ensure that your code is shared in a way that follows best practice and facilitates reproducibility and reuse.

Answer:

 All the code is available at a github repository (https://github.com/ldmnch/twitter_hate_speech) which will be made public upon acceptance. 

Please remove your figures from within your manuscript file, leaving only the individual TIFF/EPS image files, uploaded separately. These will be automatically included in the reviewers’ PDF. 

Answer:

 All figures were removed and uploaded as TIFF

Please upload a copy of Table S1 to S5 and Figures S1 to S6 to which you refer in your text. Please amend the file type to 'Supporting Information'. If the Supplementary file is no longer to be included as part of the submission please remove all reference to it within the text. 

Answer:

 Sorry for the confusion. We have revised all the supplementary material references in the texts and upload it accordingly

Please include captions for your Supporting Information files at the end of your manuscript, and update any in-text citations to match accordingly. Please see our Supporting Information guidelines for more information: http://journals.plos.org/plosone/s/supporting-information.

 Answer:

All captions were added

Reviewer #1:

While the paper provides a comprehensive review of existing literature, it lacks a clear theoretical framework to guide the analysis. Incorporating theoretical perspectives from communication studies, sociology, or critical discourse analysis could enhance the paper's theoretical depth and analytical rigor. Moreover, the paper briefly mentions methodological approaches such as text mining, machine learning, and topic modeling but lacks detailed descriptions of the methodologies employed in the studies reviewed. Providing more information about the specific methods used in hate speech detection and analysis would help readers evaluate the validity and reliability of the findings.

Answer:

We thank the reviewer for this comment. We agree that an extended theoretical framework was needed. For this, we expanded the introduction and added a section 2. Theoretical Framework and divided the introduction to add a section 3. Related work. 

While the paper acknowledges the contextual nature of hate speech, it could benefit from a more in-depth exploration of regional variations, particularly in the Latin American context. Providing case studies or examples of hate speech dynamics in specific regions would enrich the analysis and facilitate a more nuanced understanding of the phenomenon.

Answer:

We agree that a better contextualization of hate-speech in Latin America was needed. We extended the introduction to add this context.

Hate speech detection and analysis raise ethical concerns related to privacy, bias, and potential harm to vulnerable communities. The paper could benefit from a discussion of ethical considerations inherent in hate speech research and how researchers address these concerns in their methodologies and practices.

Answer:

We agree that automatic hate speech raises moral concerns that need to be acknowledge. Following Reviewer #2 comments, we added a dedicated section 4.4. on the limitations of our methodology.

Reviewer #2:

A “Related work” section is completely missing and without it it’s very difficult to compare this work with state of the art methods and consequently the impact of your scientific contributions. Please integrate such section in order to report which results and findings have been obtained in other existing works and how eventually your work differs from existing ones.

Answer:

We agree with the reviewer that adding a Related work section can help to better contextualise our contribution. We added this section to the current manuscript.

The use of advanced commercial or open-source large language models (LLMs) (e.g., ChatGPT, Mixtral, etc.) as zero-shot classifiers for automatic classification of aspects of interest (sentiment, hate speech, etc.) should be considered in your work as a better alternative to overcome some of the limitations of the ad-hoc models used in this study. In particular, such LLMs are generally very capable of analyzing text correctly and contextualizing input text, even in cases where content originates from social media (e.g., slang, emojis, etc.). In addition, such tools are able to properly interpret a text by determining if some hate speech is against someone specific, not just by considering if in the text there are specific keywords or user mentions that coud confuse a basic NLP method based on traditional machine learning. Therefore, the use of such tools could provide more accurate results, ultimately improving the overall quality of your findings.

Answer:

We thank the reviewer for this suggestion. Indeed, using a LLM allowed us to further refine the results. Although our experiments showed that pysentimiento (Pérez JM, Giudici JC, Luque F. pysentimiento: A Python Toolkit for Sentiment Analysis and SocialNLP tasks. arXiv; 2021) has a better performance to detect hate speech for this very specific linguistic context, our qualitative analysis also showed that some of the hate speech detected, although it was real hate speech, was not always targeting the mentioned candidate. In some cases, for example, it was hate speech that was supporting the original hateful message of a candidate. To complement the hate-speech detection and the qualitative analysis, we used chat GPT to fine tune our results with the prediction of the directionality of the hate speech (to detect if the candidates where or not the target of the message). These results were added at the end of section 5.3. together with the new figures 4 and 5. 

To enhance content organization and clarity, I recommend dedicating a separate section to discussing potential limitations of the methodology applied in this study. This approach will enable readers to focus more easily on the research design and methods used, while also reflecting on aspects related to the generalizability of the findings.

Answer:

We agree that a dedicated section for the limitations simplifies the reading of the methods. We compiled all the discussion on limitations on the new section 4.4 and the end of Data & Methods.

Unable to find in the article some referenced tables and/or figures (e.g. Table S5, Figure S6, etc.).

Answer:

Sorry for this omission, we submitted the supplementary materials in the current version.

---

## [Decision Letter · Decision Letter 1]

11 Dec 2024

PONE-D-24-03284R1Gender biases and hate speech: promoters and targets in the Argentinean political contextPLOS ONE

Dear Dr. Domenech Burin,

Thank you for submitting your manuscript to PLOS ONE. After careful consideration, we feel that it has satisfied our scientific requirements for publication.

However, our editorial team has noted the following text in the manuscript:

"Also, in our random sample of 15 cases we retrieved five tweets from the same user —'TurcoWturco'— mentioning a conspiracy theory of a cloned sister of Fernandez, which reflects on a systematic type of violence, rather than spontaneous social media comments. This user repeated the same tweet several times (each time mentioning different users), which could indicate that it is a troll."

It is not clear that naming the specific user is necessary for the accuracy or replicability of the study. We would therefore request that you please revise the submission to remove the user name, or include a clear scientific rationale for its inclusion.

We look forward to receiving your revised manuscript.  Jan 25 2025 11:59PM

Kind regards,

Vanessa Carels

Staff Editor

on behalf of

Stefano Cresci

Academic Editor

PLOS ONE

Journal Requirements:

Additional Editor Comments:

Thank you for sending this revised version of you r manuscript.

The reviewers are now happy with the edits and recommend publication.

Congratulations.

Reviewers' comments:

Reviewer's Responses to Questions

**Comments to the Author**

1. If the authors have adequately addressed your comments raised in a previous round of review and you feel that this manuscript is now acceptable for publication, you may indicate that here to bypass the “Comments to the Author” section, enter your conflict of interest statement in the “Confidential to Editor” section, and submit your "Accept" recommendation.

Reviewer #2: All comments have been addressed

2. Is the manuscript technically sound, and do the data support the conclusions?

Reviewer #2: Yes

3. Has the statistical analysis been performed appropriately and rigorously? 

Reviewer #2: N/A

4. Have the authors made all data underlying the findings in their manuscript fully available?

Reviewer #2: Yes

5. Is the manuscript presented in an intelligible fashion and written in standard English?

Reviewer #2: Yes

6. Review Comments to the Author

Reviewer #2: I appreciate the fact that the authors have tried to include the suggestions proposed by reviewers to improve the quality of the paper. After the revisions, the paper is more comprehensible, and the scientific contribution it provides is now at an appropriate level for publication. However, after reading the revised paper I have some other suggestions in order to improve the work. The new section “Related Work,” although it includes a substantial amount of relevant studies, does not clearly specify how the proposed work improves upon the state of the art. Neither in the “Related Work” section nor in the “Introduction” is there a discussion of the current limitations of previous studies on hate speech in the context of Argentina or South America, and how the proposed study addresses these gaps. For example, has gender bias in hate speech ever been studied in South America or Argentina? If so, what has been done, and what are the limitations of such studies? Explicitly addressing these questions could help clarify the knowledge gap in the current literature that your work seeks to fill. Although not mandatory (I am not asking an additional review), I suggest the authors to also cover these points in the final version of the article, to improve readability and motivations of their work.

7. PLOS authors have the option to publish the peer review history of their article (what does this mean?). If published, this will include your full peer review and any attached files.

Reviewer #2: No

---

## [Author Response · Author response to Decision Letter 1]

12 Dec 2024

Editorial team:

1. However, our editorial team has noted the following text in the manuscript:

"Also, in our random sample of 15 cases we retrieved five tweets from the same user —'TurcoWturco'— mentioning a conspiracy theory of a cloned sister of Fernandez, which reflects on a systematic type of violence, rather than spontaneous social media comments. This user repeated the same tweet several times (each time mentioning different users), which could indicate that it is a troll."

It is not clear that naming the specific user is necessary for the accuracy or replicability of the study. We would therefore request that you please revise the submission to remove the user name, or include a clear scientific rationale for its inclusion.

Answer:

We thank the editorial team for this comment. Indeed, highlighting the name of a single user is unnecessary and detrimental for the article. We therefore removed the user name.

Reviewer #2:

I appreciate the fact that the authors have tried to include the suggestions proposed by reviewers to improve the quality of the paper. After the revisions, the paper is more comprehensible, and the scientific contribution it provides is now at an appropriate level for publication. However, after reading the revised paper I have some other suggestions in order to improve the work. The new section “Related Work,” although it includes a substantial amount of relevant studies, does not clearly specify how the proposed work improves upon the state of the art. Neither in the “Related Work” section nor in the “Introduction” is there a discussion of the current limitations of previous studies on hate speech in the context of Argentina or South America, and how the proposed study addresses these gaps. For example, has gender bias in hate speech ever been studied in South America or Argentina? If so, what has been done, and what are the limitations of such studies? Explicitly addressing these questions could help clarify the knowledge gap in the current literature that your work seeks to fill. Although not mandatory (I am not asking an additional review), I suggest the authors to also cover these points in the final version of the article, to improve readability and motivations of their work.

Answer:

We thank the reviewer for this comment. We agree that explicitly addressing these questions clarify the knowledge gap in literature and the relevance for our work in the field. To address this, we extended a paragraph in the introduction, to include the research gap before the research questions.

---

## [Editor Report · Decision Letter 2]

20 Dec 2024

Gender biases and hate speech: promoters and targets in the Argentinean political context

PONE-D-24-03284R2

Dear Dr. Domenech Burin,

We’re pleased to inform you that your manuscript has been judged scientifically suitable for publication and will be formally accepted for publication once it meets all outstanding technical requirements.

Kind regards,

Stefano Cresci

Academic Editor

PLOS ONE
---

## [Editor Report · Acceptance letter]

14 Jan 2025

PONE-D-24-03284R2 

PLOS ONE

Dear Dr. Domenech Burin, 

I'm pleased to inform you that your manuscript has been deemed suitable for publication in PLOS ONE. Congratulations! Your manuscript is now being handed over to our production team.

Kind regards, 

on behalf of

Dr. Stefano Cresci 

Academic Editor

PLOS ONE